# Genetic and compound screens uncover factors modulating cancer cell response to indisulam

Ziva Pogacar[1],*, Kelvin Groot[1],*, Fleur Jochems[1], Matheus Dos Santos Dias[1], Antonio Mulero-Sánchez[1], Ben Morris[2], Mieke Roosen[1], Leyma Wardak[1], Giulia De Conti[1], Arno Velds[3], Cor Lieftink[2], Bram Thijssen[1], Roderick L Beijersbergen[1,2,3], René Bernards[1], Rodrigo Leite de Oliveira[1]

Discovering biomarkers of drug response and finding powerful drug combinations can support the reuse of previously abandoned cancer drugs in the clinic. Indisulam is an abandoned drug that acts as a molecular glue, inducing degradation of splicing factor RBM39 through interaction with CRL4[DCAF15]. Here, we performed genetic and compound screens to uncover factors mediating indisulam sensitivity and resistance. First, a dropout CRISPR screen identified SRPK1 loss as a synthetic lethal interaction with indisulam that can be exploited therapeutically by the SRPK1 inhibitor SPHINX31. Moreover, a CRISPR resistance screen identified components of the degradation complex that mediate resistance to indisulam: DCAF15, DDA1, and CAND1. Last, we show that cancer cells readily acquire spontaneous resistance to indisulam. Upon acquiring indisulam resistance, pancreatic cancer (Panc10.05) cells still degrade RBM39 and are vulnerable to BCL-xL inhibition. The better understanding of the factors that influence the response to indisulam can assist rational reuse of this drug in the clinic.

## Introduction

Personalised anti-cancer therapy is limited by high costs of drug development (Workman et al, 2017; Schlander et al, 2021). One strategy to lower the costs is to reuse compounds already tested in the clinical setting but that were abandoned because of lack of single agent activity. As most abandoned drugs are no longer patent protected, their reuse will be more affordable. The understanding of the molecular mechanism of action of those compounds allows identification of biomarkers of response and the discovery of combination treatments. This knowledge might lead to a rational strategy to reuse previously abandoned drugs.

One example of a previously abandoned drug is indisulam, which was first described as a sulfonamide with anti-cancer activity with an unknown mechanism of action (Owa et al, 1999; Fukuoka et al, 2001). Indisulam was tested in multiple clinical trials, where it was proven to be safe and well tolerated, but had limited efficacy (clinical responses and stable disease in 17–35% of advanced stage cancer patients) (Punt et al, 2001; Raymond et al, 2002; Dittrich et al, 2003; Terret et al, 2003; Haddad et al, 2004; Smyth et al, 2005; Yamada et al, 2005; Talbot et al, 2007; Assi et al, 2018). Because of the modest response rates, the further clinical development of indisulam was halted. However, expired patent protection and the discovery of indisulam's molecular mechanism of action as a molecular glue may facilitate the re-introduction into clinical development (Han et al, 2017; Uehara et al, 2017).

Molecular glues and proteolysis targeting chimeras (PROTACs) are a novel type of compounds that exploit the endogenous ubiquitin-proteasome system to induce targeted protein degradation of neo-substrates (Scholes et al, 2021). As a molecular glue, indisulam facilitates the interaction between RNA-binding motif protein 39 (RBM39) and DDB1 and CUL4-associated factor 15 (DCAF15) in the cullin-RING E3 ubiquitin ligase 4 complex (CRL4[DCAF15]) resulting in ubiquitination and proteasomal degradation of RBM39 (Han et al, 2017; Uehara et al, 2017). The activity of cullin-RING ubiquitin ligases (CRLs) is regulated by post-translational modification with NEDD8 (Ohh et al, 2002) which leads to the transfer of ubiquitin to a substrate. Furthermore, the exchange factor cullin associated and neddylation dissociated 1 (CAND1) allows the exchange of the substrate receptor of de-neddylated CRL and increases the diversity of substrates that can be degraded (Liu et al, 2002). Indisulam treatment leads to the interaction between CRL4[DCAF15] and RBM39, as recently demonstrated by the resolved structure of the interacting complex (Bussiere et al, 2020). RBM39 is a splicing factor involved in early spliceosome assembly (Stepanyuk et al, 2016) and its loss leads to the accumulation of splicing errors and cytotoxicity (Wang et al, 2019; Ting et al, 2019; Han et al, 2017).

[1]Division of Molecular Carcinogenesis, Oncode Institute, The Netherlands Cancer Institute, Amsterdam, The Netherlands [2]The Netherlands Cancer Institute Robotics and Screening Center, The Netherlands Cancer Institute, Amsterdam, The Netherlands [3]Genomics Core Facility, The Netherlands Cancer Institute, Amsterdam, The Netherlands

Correspondence: r.bernards@nki.nl; r.ld.oliveira@amsterdamumc.nl
Rodrigo Leite de Oliveira's present address is CRISPR Expertise Center, Cancer Center Amsterdam, Amsterdam University Medical Center, Amsterdam, The Netherlands.
*Ziva Pogacar and Kelvin Groot contributed equally to this work.

Understanding drug resistance mechanisms can further aid in biomarker discovery and help guide combination treatment. It has been described that point mutations in RBM39 prevent the interaction with DCAF15 leading to resistance of HCT-116 colon cancer cells to indisulam (Han et al, 2017; Ting et al, 2019). Similarly, knock-out of *DCAF15* prevents RBM39 degradation and confers resistance (Han et al, 2017). Recently, *CAND1* loss has been described to induce resistance to multiple degraders, including indisulam (Mayor-Ruiz et al, 2019). However, the clinical significance of these resistance mechanisms is still unclear. Here we use functional genetic and compound screens to identify genes that modulate the response to indisulam.

# Results

### SRPK1 loss is synthetic lethal with indisulam

In an effort to re-position indisulam for treatment of solid tumors, we first characterized the response to indisulam in a panel of solid tumor cell lines from different tissue types (pancreas, lung, breast, colon). We observed a range of responses of solid cancer cell lines to indisulam, with some cell lines being very sensitive to indisulam (colon cancer cell line HCT-116), others moderately sensitive (e.g., A549 lung cancer cell line) and some resistant up to 2 $\mu$M of indisulam (e.g., SUM159 breast cancer cell line) (Fig 1A). Because splicing factor RBM39 is the molecular target of indisulam, we then characterized the dynamics of RBM39 degradation in these cell lines. The levels of residual RBM39 after 72 h of indisulam treatment correlated with the sensitivity of the cell line. Sensitive cell lines showed no residual RBM39 after 72 h whereas moderately sensitive cell lines and resistant cell lines still retained detectable RBM39 levels (Fig 1B).

The variable response to indisulam between solid cancer cell lines suggests that cell-intrinsic factors mediate sensitivity to indisulam. In addition, because many cell lines do not respond to indisulam monotherapy there is a need to identify possible indisulam combination treatments. To address this, we performed a synthetic lethality CRISPR screen in the moderately sensitive line A549 using an sgRNA library targeting the human kinome (Wang et al, 2018). The cells were cultured for 10 d in the presence or absence of 0.35 $\mu$M of indisulam. After this, sgRNAs were recovered by PCR and the abundance of gRNAs in the two conditions were determined by NGS as described previously (Evers et al, 2016). When we analyzed the relative abundance of sgRNAs in the indisulam-treated condition compared with untreated, we observed a depletion of sgRNAs targeting *SRPK1* (Fig 1C). SRPK1 is a serine/arginine protein kinase which acts as a regulator of constitutive and alternative splicing (Wang et al, 1999). To validate the synthetic lethal interaction between indisulam and *SRPK1* loss, we generated single cell *SRPK1* knock-out clones in A549, SUM159, and DLD1 cells (Figs 1D and S1A). *SRPK1* knock-out clones were more sensitive to indisulam than control cells in all cell lines, confirming the result of the CRISPR screen (Figs 1E–H and S1B and C). Taken together, we show that loss of *SRPK1* is synthetic lethal with indisulam treatment in multiple cancer cell lines.

### Combination of indisulam and SRPK1 inhibitor impairs cell proliferation

Next, we tested a specific SRPK1 inhibitor SPHINX31 and observed that combination of SPHINX31 and indisulam impaired proliferation

of A549 (Fig 2A) as well as H2122 and SUM159 (Fig S2A). Furthermore, we observed an increase in apoptosis measured by caspase 3/7 activity in cells treated with the combination (Fig S2B and C). To investigate if the combination of indisulam and SPHINX31 is synergistic or additive we performed a viability experiment using a matrix of concentrations and calculated the Bliss synergy score. A Bliss score above 10 indicates synergy. We observed that the combination of indisulam and SPHINX31 is synergistic in A549 and SUM159, but less in H2122 (Fig 2B). We noticed that the cytotoxic effect of indisulam combined with SPHINX31 was more potent than the genetic knock-out of *SRPK1* combined with indisulam. To investigate potential off-target effects of SPHINX31 we performed a viability experiment using a matrix of concentrations of indisulam and SPHINX31 in *SRPK1* knock-out clones and control cells. We noticed that there was still synergy in clone #2.1 and less in clone #2.2. This indicates potential off-target activity of SPHINX31, which is not surprising because it was reported to also target CLK1 and SRSF2 (Batson et al, 2017) (Fig S2D). As CLK1 interacts with SRPK1 to facilitate spliceosome assembly, inhibiting both proteins might explain the observed synergy in *SRPK1* knock-out clones (Aubol et al, 2016).

To study if the effect of indisulam combined with SPHINX31 is mediated by RBM39 loss, we used shRNAs to knock down *RBM39*. Because *RBM39* is an essential gene, only partial knockdown is achievable without compromising cell viability (Fig 2C and D). *RBM39* knock-down cells showed increased response to SPHINX31 (Fig 2E and F), consistent with the notion that indisulam-induced RBM39 degradation sensitized to SPHINX31.

Because both RBM39 and SRPK1 are involved in splicing, we asked whether the synergistic effect between indisulam and SPHINX31 can be explained by an increased amount of splicing errors. We treated A549 cells with indisulam, SPHINX31, and the combination for 24 h and quantified splicing errors using transcriptome analysis. Treatment with indisulam increased splicing errors, most notably skipped exons (Fig 2G). There were splicing errors detected in SPHINX31-treated cells, but at a much lower frequency. Interestingly, the combination of indisulam and SPHINX31 increased the number of skipped exons beyond what would be expected from the sum of the single treatments. This could indicate a threshold of splicing errors that is compatible with viability. To study the long term effects of the indisulam and SPHINX31 combination we performed a long-term colony-formation assay in A549, H2122, and SUM159 cells (Figs 2H and I and S2E and F). Even though all cell lines acquired resistance to indisulam after 2–4 wk of treatment, combination of indisulam and SPHNIX31 prevented acquired resistance in all three cell lines.

Taken together, we show that synergy between indisulam and SPHINX31 is mediated by indisulam induced RBM39 degradation and that combination treatment prevents acquired resistance to indisulam.

### Resistance to indisulam through CAND1 loss and reduced RBM39 degradation

To understand which factors mediate indisulam resistance, we performed a genome-wide resistance screen in A549 cells treated with indisulam. The cells were treated with 3 $\mu$M of indisulam or

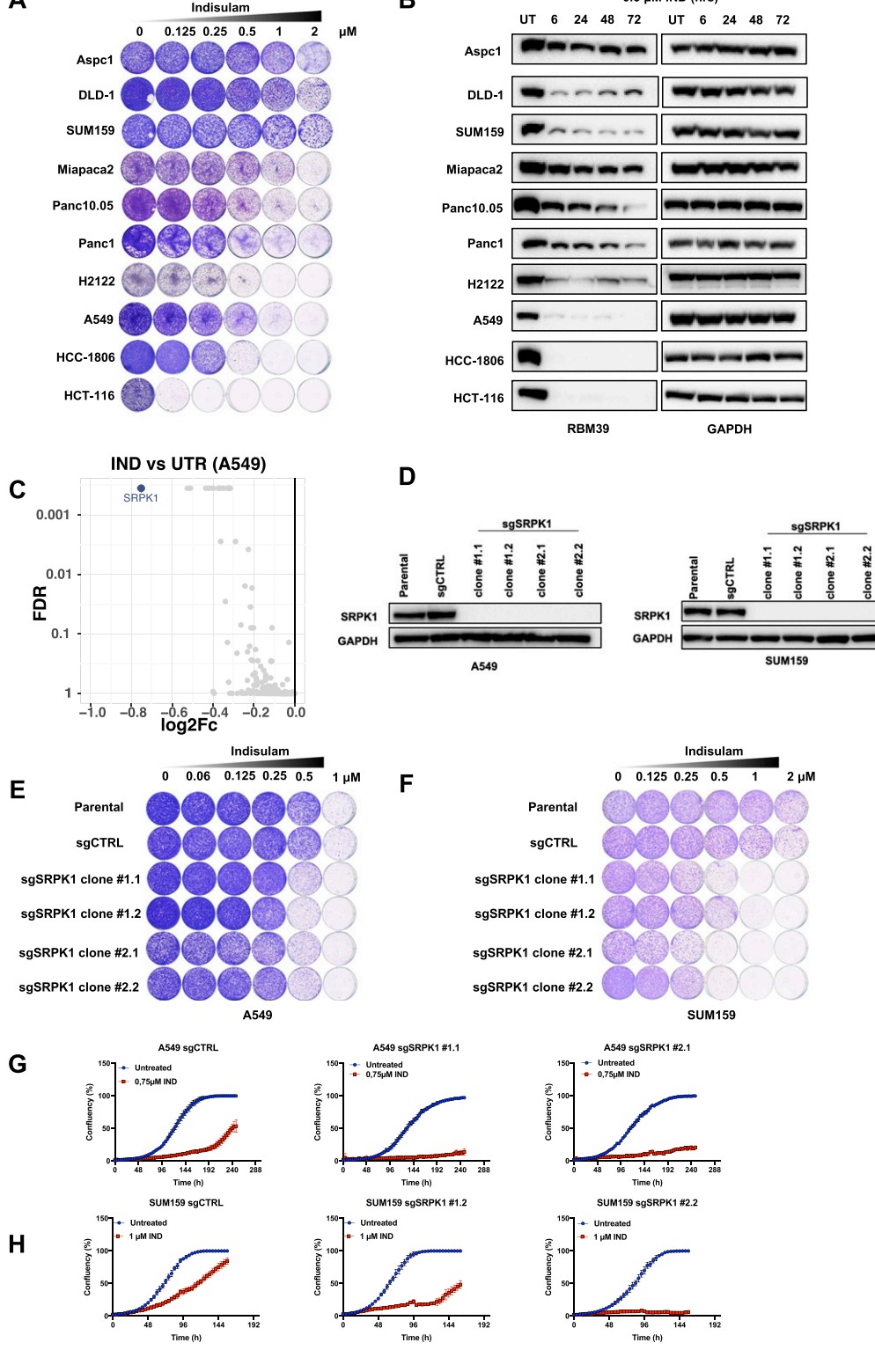

**Figure 1. Dropout screen identifies SRPK1 as synthetic lethal with indisulam treatment.**
**(A)** Long-term colony-formation assays of Aspc1, DLD-1, SUM159, Miapaca2, Panc10.05, Panc1, A549, H2122, HCC-1806, and HCT-116. Cells were treated with indicated doses of indisulam for 8–11 d. **(B)** Western blot analysis of RBM39 levels in Aspc1, DLD-1, SUM159, Miapaca2, Panc10.05, Panc1, A549, H2122, HCC-1806, and HCT-116 cells treated with 0.5 $\mu$M of indisulam for the indicated time periods. GAPDH was used as a loading control. **(C)** Dropout CRISPR screen was performed in A549 treated with 0.35 $\mu$M indisulam. Volcano plot of indisulam-treated samples compared with untreated. X axis shows log$_2$ fold change of normalized read counts and Y axis shows false discovery rate (FDR). Each dot represents an individual gene and *SRPK1* is highlighted. **(D)** Western blot analysis of SRPK1 levels in A549 and SUM159 SRPK1 knock-out clones and control cells. Clones were generated from two independent sgRNAs. GAPDH was used as a loading control. **(E)** Long-term colony-formation assay of A549 cells. A549 *SRPK1* knock-out clones and control cells were treated with indicated doses of indisulam for 10 d. **(F)** Long-term colony-formation assay of SUM159 cells. SUM159 *SRPK1* knock-out clones and control cells were treated with indicated doses of indisulam for 7 d. **(G)** Proliferation assay of A549 control and sgSRPK1 cells treated with 0.75 $\mu$M indisulam. One clone per sgRNA is shown. Mean of three technical replicates is shown and error bars indicate SD. **(H)** Proliferation assay of SUM159 control and sgSRPK1 cells treated with 1 $\mu$M indisulam. One clone per sgRNA is shown. Mean of three technical replicates is shown and error bars indicate SD.
Source data are available for this figure.

control media for 3 wk. After this, we identified the enriched sgRNAs between the two conditions by NGS of the recovered gRNAs. When comparing the treated versus untreated condition we observed enrichment of sgRNAs targeting *DCAF15*, *DDA1*, and *CAND1* (Fig 3A).

Both DCAF15 and DDA1 are part of the CRL complex and their loss impairs the degradation of RBM39 as previously described (Han et al, 2017; Mayor-Ruiz et al, 2019). We therefore focused on validating *CAND1* as its function in indisulam resistance was less understood

at the time. CAND1 acts as a substrate receptor exchange factor regulating CRL complex activity (Liu et al, 2018; Reichermeier et al, 2020). We knocked out *CAND1* in A549 cells and observed decreased sensitivity to indisulam in knock-out cells compared with control cells (Figs 3B and C and S3A). We confirmed the resistance caused by *CAND1* knockout in another moderately sensitive cell line, Panc10.05 (Figs 3E and S3B). We then investigated RBM39 degradation in *CAND1* knock-out cells and observed reduced degradation of RBM39 compared with control cells (Fig 3D and F). On the other hand, in the sensitive cell line HCT-116 we observed much less RBM39 stabilisation and there was no increase in resistance upon *CAND1* knock-out (Fig S3C–E). This suggests that the levels of RBM39 resulting from *CAND1* loss are not high enough to sustain HCT-116 cell viability upon indisulam treatment.

Next, we asked if a further increase in RBM39 stabilisation would lead to indisulam resistance in HCT-116 cells. We made use of MLN4924, a neddylation inhibitor which inhibits the NEDD8 activating E1 enzyme (NAE) and prevents the activation of CRLs (Fig 3G). Treatment with MLN4924 reduced CUL4A neddylation and prevented RBM39 degradation in both HCT-116, as well as in the moderately sensitive cell line A549 (Fig 3H). In addition, we used the proteasome inhibitor MG-132 which prevents RBM39 degradation, but does not impair neddylation. Increasing the concentration of MLN4924 resulted in increased levels of RBM39 both in HCT-116 and A549 (Fig 3I). Notably, in the less sensitive cell line A549 a higher concentration of MLN429 still leads to less RBM39 stabilisation compared with HCT-116. Next, we treated HCT-116 and A549 cells with a combination of indisulam and MLN4924. We observed a rescue of indisulam toxicity when adding MLN4924 in HCT-116, but not in A549 cells (Fig 3J). In addition, we performed a synergy analysis and observed antagonism of indisulam and MLN4924 in HCT-116 but not in A549 (Fig S3F and G). As A549 cells are less sensitive to indisulam, a higher concentration of MLN4924 is required to stabilize RBM39. Because MLN4924 becomes toxic at higher concentrations, there is no rescue of cell viability in A549. This is even more apparent in the synergy analysis, as it becomes clear that there is a much smaller window to detect antagonism in A549 (Fig S3G). These data indicate that increasing RBM39 levels either by *CAND1* knock-out or inhibition of neddylation results in indisulam resistance.

### Cells with acquired resistance to indisulam are vulnerable to BCL-XL inhibition

In addition to loss of function mutations, gradual adaptation to drug treatment can also lead to drug resistance. To study spontaneous resistance to indisulam, we cultured various cell lines with increasing concentrations of indisulam. We observed that all tested cell lines acquired resistance to indisulam after 3 mo of culture in the presence of the drug (Fig 4A and B). Next, we asked if resistant cells were still able to degrade RBM39. We observed a large difference in RBM39 degradation between cell lines (Fig 4C and D). Resistant HCT-116 cells showed an increase in RBM39 in the presence of indisulam, whereas HCC-1806 and A549 cells still showed some degradation of RBM39 in the presence of indisulam. Next, we tested if the differences in RBM39 levels could be explained by loss of *CAND1* in resistant cells. We did not observe any

changes of CAND1 levels between resistant and parental cells (Fig S4A) indicating that the differences in RBM39 are likely CAND1 independent. Interestingly, Panc10.05 cells show a strong reduction in RBM39 levels without impairing cell viability. Because this indicates an RBM39 independent resistance mechanism, we characterized this resistance further. As degradation of RBM39 results in the accumulation of splicing errors we first asked whether resistant Panc10.05 cells that degrade RBM39 still accumulate splicing errors. Transcriptome analysis of parental and resistant Panc10.05 cells treated with indisulam revealed that resistant cells had lower levels of splicing errors than control parental cells (Fig 4E). This could indicate that lowering the number of splicing errors allows the resistant cells to survive.

Next, we studied if Panc10.05 cells resistant to indisulam also acquired a therapeutically exploitable vulnerability. We made use of a compound library consisting of 164 anti-cancer compounds (Table S1). After screening the compounds on parental and resistant Panc10.05 cells, we identified a list of candidate compounds that had greater impact on viability of resistant than parental cells based on the difference in AUC (Fig 4F). Four of the highest scoring compounds were rapamycin, prexasertib, A-1155463, and ABT-263. After the secondary screen, we focused on validation of A-1155463 and ABT-263 and excluded compounds with unclear dose response curves (rapamycin) or those that showed very small difference between parental and resistant cells (prexasertib). As we validated the effect of inhibitors targeting the anti-apoptotic protein BCL-xL on parental and resistant cells, we observed that indisulam-resistant Panc10.05 cells were more sensitive to both ABT-263 (BCL-2, BCL-xL, and BCL-W inhibitor) and A-1155463 (BCL-xL inhibitor) than parental control cells (Fig 4G). On the other hand, indisulam-resistant A549 and HCC1806 did not show an increased sensitivity to ABT-263 and A-1155463 compared with parental cells, indicating that this might be cell line specific or specific to resistant cell lines with low RBM39 levels (Fig S4B).

ABT-263 and A-1155463 are BH3 mimetics as they mimic pro-apoptotic BH3-domain only proteins in targeting anti-apoptotic proteins. Because both ABT-263 and A-1155463 target BCL-xL, we checked the levels of BCL-xL in parental and resistant cells. There was a modest increase in BCL-xL protein both in parental cells treated with indisulam as well as resistant cells treated with indisulam (Fig 4H). Apoptosis is mostly regulated on the post-translational level and is highly dependent on the balance of anti- and pro-apoptotic signals (Giam et al, 2008). To understand specific apoptotic dependencies of parental and indisulam-resistant cells we made use of BH3 profiling, an assay that measures mitochondrial outer membrane permeabilization in response to BH3 peptides derived from BH3-domain only proteins (Ryan & Letai, 2013). We treated the parental and indisulam-resistant Panc10.05 cells with various BH3 peptides and inhibitors and measured cytochrome c release using flow cytometry. Treatment with BAD, HRK, as well as another BCL-xL inhibitor A-1331852 and ABT-263 triggered a stronger cytochrome C release in resistant cells compared with parental control cells (Fig 4I). This indicates a higher dependency of indisulam-resistant Panc10.05 cells on BCL-xL, which could contribute to the resistance phenotype.

We then asked if we can exploit the dependency of resistant cells on BCL-xL to prevent the development of the resistance. To this end,

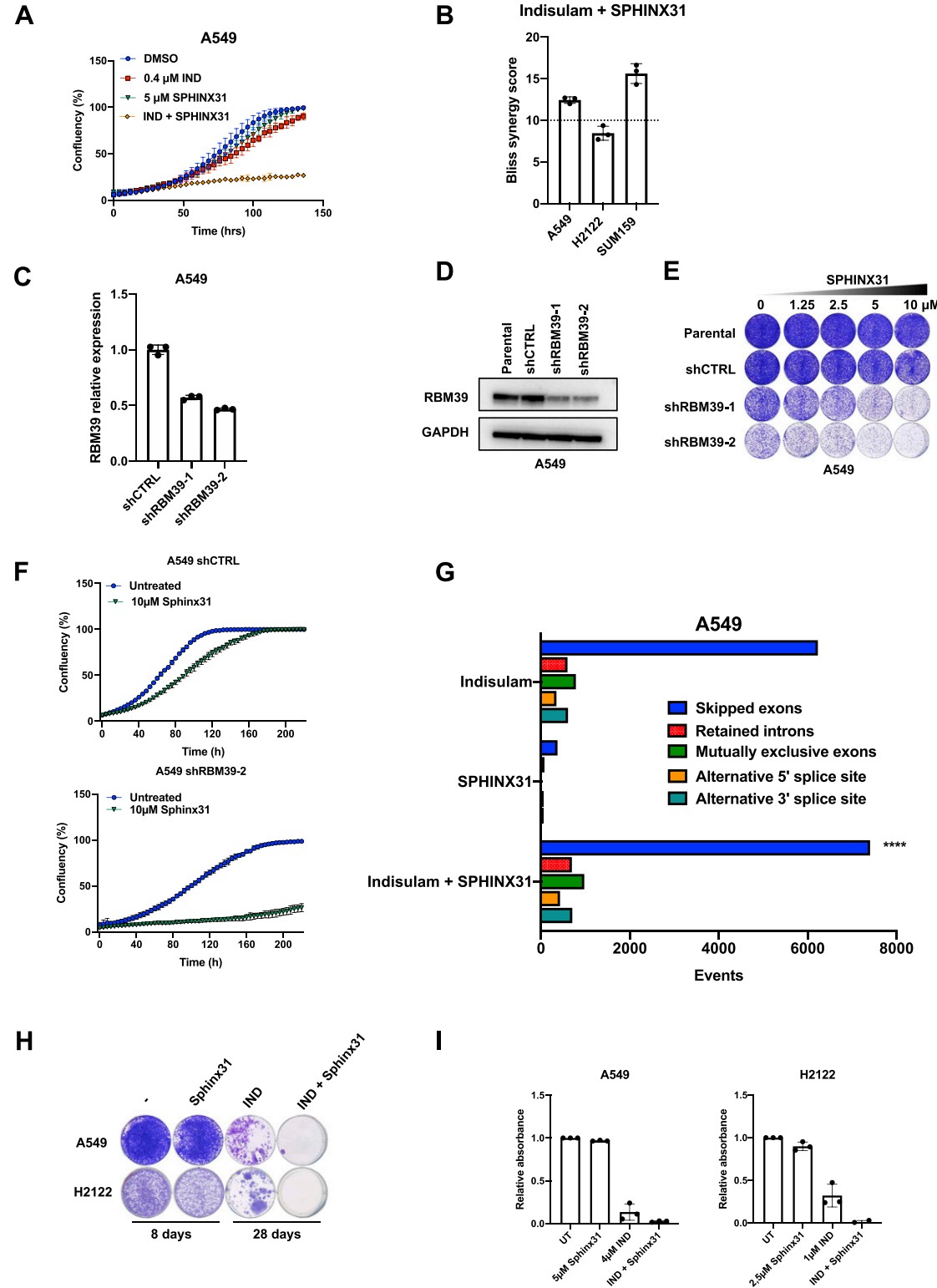

**Figure 2. Combination of indisulam and SPRK1 inhibitor impairs cell proliferation and prevents acquired resistance to indisulam.**
**(A)** Proliferation assay of A549 cells treated with 0.4 $\mu$M indisulam, 5 $\mu$M SPHINX31, and the combination. Mean of three technical replicates is shown and error bars indicate SD. **(B)** Drug synergy analysis of a 6-d treatment with indisulam in combination with SPHINX31 in A549, H2122, and SUM159 cells. Bliss synergy score cut-off of 10 is shown, indicating likely synergy. Mean of three biological replicates is shown and error bars indicate SD. **(C)** qPCR analysis of RBM39 normalized to housekeeping gene RPL13 in A549. Mean of three technical replicates is shown and error bars indicate SD. **(D)** Western blot analysis of RBM39 levels in shRBM39 and control A549 cells. GAPDH was used as a loading control. **(E)** Long-term colony-formation assay of A549. shRBM39 and control cells were treated with indicated doses of indisulam for 10 d.

we treated parental Panc10.05 cells with ABT-263, A-1155463, indisulam and the combinations. As expected, parental cells were not sensitive to monotherapy of either ABT-263 or A-1155463. Even though indisulam is initially effective, cells acquired resistance after 4 wk of culture on indisulam. However, the combination of indisulam with ABT-263 and A-1155463 completely prevented the development of resistance in Panc10.05 cells (Fig 4J and K). We then asked whether other pancreatic cancer cell lines treated with indisulam also show a dependency on BCL-xL. We treated Panc1, Miapaca2, and Aspc1 cells with ABT-263, A-1155463, and indisulam (Fig S4C and D). All cell lines acquired resistance to indisulam after 4 wk. The combination of ABT-263 and A-1155463 prevented resistance in Aspc1 cells and Panc1 cells. On the other hand, in Miapaca2 cell line we observed a reduction in resistance after treating the cells with the combination of indisulam and ABT-263, but not A-1155463. This might indicate that this cell line is more dependent on BCL-2 or BCL-w rather than on BCL-xL. Furthermore, we did not observe any major differences in BCL-2 and BCL-xL abundance upon indisulam treatment in Miapaca, Aspc1 and Panc1 (Fig S4E). Taken together, there seem to be different dependencies on anti-apoptotic proteins between cell lines treated with indisulam. However, in some cases combining indisulam with a BCL-xL inhibitor can prevent the development of spontaneous resistance.

# Discussion

Drug repurposing is an attractive strategy that can contribute to affordable healthcare (Zhang et al, 2020). Here, we suggest that the previously abandoned anti-cancer compound indisulam has great potential to be reused because of expired patent protection, favourable safety profile in the clinic and a recently described molecular mechanism of action. Biomarkers of response and new combination treatments are instrumental for future clinical development of this drug. A great tool for both biomarker and combination treatment discovery are functional genetic screens (Mulero-Sánchez et al, 2019). Here, we identify a synthetic lethal interaction with indisulam as well as resistance mechanisms to indisulam using CRISPR screens.

We show that the response to indisulam in solid cancer cell lines is variable, which is in line with the response rate in clinical trials (Punt et al, 2001; Raymond et al, 2002; Dittrich et al, 2003; Terret et al, 2003; Haddad et al, 2004; Smyth et al, 2005; Yamada et al, 2005; Talbot et al, 2007; Assi et al, 2018). Furthermore, the in vitro response seems to correlate with the residual RBM39 levels after indisulam treatment. Interestingly, RBM39 degradation was described as a biomarker of indisulam response in acute myeloid leukemia and DCAF15 levels were shown to correlate with indisulam response in

hematopoietic and lymphoid cancers (Han et al, 2017; Hsiehchen et al, 2020). As this correlation was not observed in solid cancers there might be other factors contributing to tissue specificity of sensitivity and resistance to indisulam.

To further explore the use of indisulam in solid tumors, we performed a dropout CRISPR screen and identified loss of *SRPK1* as a synthetic lethal interaction with indisulam. SRPK1 is a splicing factor that phosphorylates serine and arginine-rich (SR) proteins, such as SRSF1, which leads to their activation and enables splicing (Gui et al, 1994; Colwill et al, 1996; Varjosalo et al, 2013; Aubol et al, 2016). A global proteomic analysis has shown that RBM39 is a direct target of SRPK1 (Varjosalo et al, 2013) which could explain the synthetic lethal interaction. Combination of SRPK1 inhibitor SPHINX31 and indisulam led to an increase of splicing errors. This could indicate that the cells can tolerate a certain amount of splicing errors, until a threshold is reached which leads to cytotoxicity. On the other hand, aberrant splicing of specific genes due to the combination might contribute to the synergy as well. Combining different splicing inhibitors may offer an advantage over single treatments (Bonnal et al, 2020). Furthermore, SRPK1 negative tumors might benefit from indisulam monotherapy treatment.

To anticipate resistance mechanisms to indisulam that can potentially arise in the clinic, we performed a whole genome resistance CRISPR screen. We identified two components of the CRL complex: DCAF15 and DDA1 as well as the substrate receptor exchange factor CAND1. This observation is in line with a previous screen that investigated resistance to multiple degraders (Mayor-Ruiz et al, 2019). Loss of CAND1 was described to lock the CRL complex in a hyper neddylated state which leads to auto-degradation of substrate receptors (Mayor-Ruiz et al, 2019). Curiously, both inhibition of neddylation and CAND1 loss lead to stabilisation of RBM39 levels and resistance. Similarly, spontaneously generated indisulam-resistant cells showed minor or no RBM39 degradation. The resistance in these cells may be mediated by point mutations in RBM39 that prevent its binding to CRL4$^{DCAF15}$, as described previously (Han et al, 2017; Ting et al, 2019). On the other hand, indisulam-resistant Panc10.05 cells still degraded RBM39. Because these cells also harbour less splicing errors, this could indicate a mechanism downstream of RBM39 that prevents splicing errors and allows survival. Interestingly, Panc10.05 cells depend on BCL-xL and spontaneous resistance can be prevented by co-treatment with BCL-xL inhibitors ABT-263 and A-1155463. This is in line with a previous report that showed synergy of splicing modulators and BCL-xL inhibitors (Aird et al, 2019). Combination treatment with BCL-xL inhibitors and indisulam could therefore prevent acquired resistance and lead to improved treatment success in the clinical setting.

Cancer types that harbour mutations in the spliceosome, such as hematopoietic and lymphoid malignancies, seem to be more

---

**(F)** Proliferation assay of shRBM39 and control A549 cells treated with 10 $\mu$M SPHINX31. Mean of three technical replicates is shown and error bars indicate SD.
**(G)** Quantification of splicing errors in RNA sequencing data from A549 cells treated for 24 h with 0.5 $\mu$M indisulam, 5 $\mu$M SPHINX31, and the combination. Data were analyzed based on two technical replicates and bars represent the number of events relative to untreated samples. Statistical difference was assessed by a Poisson test comparing the splicing errors in the combination to the sum of splicing errors of the individual treatments. Asterisks denote significance (****$P$ < 0.0001). **(H)** Long-term colony-formation assays of A549 and H2122. A549 were treated with 5 $\mu$M SPHINX31, 4 $\mu$M indisulam, and the combination. H2122 cells were treated with 2.5 $\mu$M SPHINX31, 1 $\mu$M indisulam, and the combination. **(I)** Quantification of long-term colony-formation assays of A549 and H2122 cells. Mean of three biological replicates is shown and error bars indicate SD.
Source data are available for this figure.

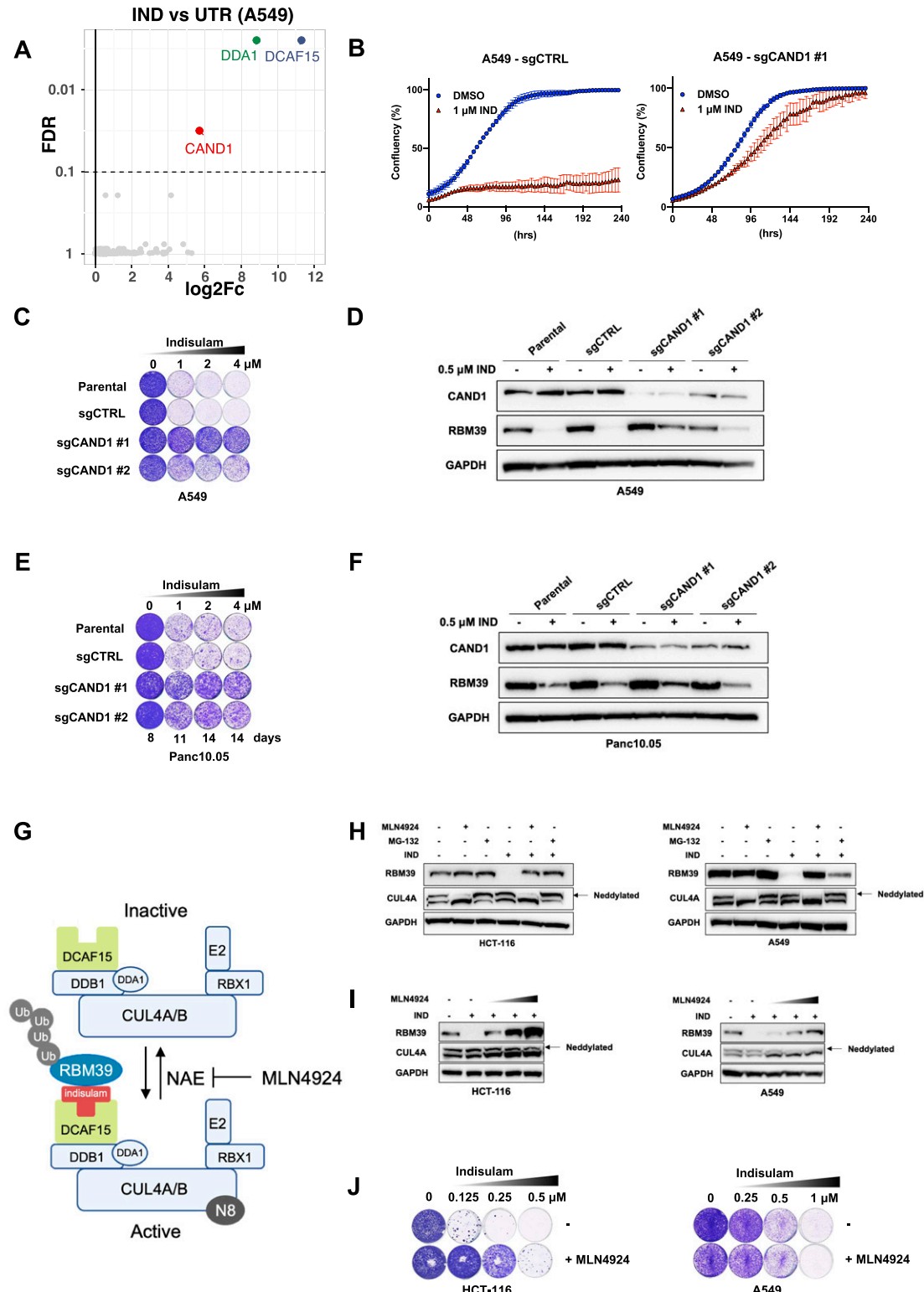

**Figure 3. Resistance to indisulam is modulated through reduced RBM39 degradation and CAND1 loss.**

**(A)** Resistance screen was performed in A549 cells treated with 3 $\mu$M indisulam. Volcano plot of indisulam-treated samples compared with untreated. X axis shows $\log_2$ fold change of normalized read counts and Y axis shows false discovery rate (FDR). Each dot represents an individual gene and hits are highlighted. **(B)** Proliferation assay of A549 control (sgCTRL) and sgCAND1 cells treated with 1 $\mu$M indisulam. Mean of three technical replicates is shown and error bars indicate SD. **(C)** Long-term colony-formation assay of A549. Wild-type, control, and two individual sgCAND1 cells were treated with indicated doses of indisulam for 10 d. **(D)** Western blot analysis of RBM39 and CAND1 in A549 cells. Wild-type, control and sgCAND1 cells were treated with 0.5 $\mu$M of indisulam for 8 d. GAPDH was used as loading control. **(E)** Long-term colony-

sensitive to indisulam (Han et al, 2017; Wang et al, 2019; Bonnal et al, 2020). Our data indicate that SRPK1 mutant solid tumors may be more sensitive to indisulam as well. However, this might be just one example of a synthetic lethal interaction and loss of other splicing factors could sensitize cells from different tissue types to indisulam as well. Furthermore, we propose that the combination of indisulam and SPHINX31 might present a better treatment strategy and that addition of either BCL-xL inhibitors or SPHINX31 might prevent acquired resistance. Recently, it has been shown that indisulam induced splicing errors can lead to neoantigen formation and that combining indisulam with immunotherapy improved treatment outcomes (Lu et al, 2021). Further understanding of the factors involved in indisulam sensitivity and resistance might help in predicting which patients would benefit from this combination treatment.

# Materials and Methods

### Cell lines

HCT-116, HCC-1806, Panc10.05, A549, Miapaca2, and H2122 cells were cultured in RPMI (Gibco) supplemented with 10% FBS (Serana) and 1% penicillin-streptomycin (Gibco). Aspc1, Panc1, and HEK293T cells were cultured in DMEM (Gibco) supplemented with 10% FBS and 1% penicillin-streptomycin (Gibco). SUM159 cells were cultured in DMEMF12 (Gibco) supplemented with 5% FBS (Serana), 1% penicillin-streptomycin (Gibco), 5 $\mu$g/ml insulin and 1 $\mu$g/ml hydrocortisone (Sigma-Aldrich). HCT-116, HCC-1806, Panc10.05, A549, Miapaca2, Aspc1, Panc1, H2122, and HEK293T were purchased from ATCC. SUM159 was a gift from Mettello Innocenti (NKI). All cell lines were maintained in a humidified incubator at 37°C and 5% $CO_2$ and were regularly tested for mycoplasma contamination using a PCR-based assay. To establish indisulam-resistant cell lines, HCT-116, HCC-1806, Panc10.05, and A549 cells were treated with increasing doses of indisulam (from 0.125 to 1 $\mu$M) for at least 2 mo. The dose of indisulam was doubled every 2 wk. At the time of the experiments, indisulam-resistant cells were cultured at 0.5 $\mu$M indisulam.

### Compounds and antibodies

Indisulam (E7070) (#201540), SPHINX31 (#555397), MLN4924 (#201924), Navitoclax (ABT-263) (#201970), and A-1155463 (#407213) were purchased from MedKoo Biosciences. MG-132 was purchased from Selleckchem. Phenylarsine oxide (PAO) was purchased from Sigma-Aldrich. All reagents were dissolved in DMSO at a stock

solution of 10 mM. A-1331852 (#HY-19741) was obtained from MedChemExpress. Antibodies against CAND1 (#8759), CUL4A (#2699), GAPDH (#5174), Bcl-2 (#2872), and BCL-xL (#2764) were purchased from Cell Signalling Technology. Antibody against RBM39 (HPA001591) was purchased from Atlas Antibodies. Antibody against SRPK1 (611072) was purchased from BD Biosciences. Antibody against vinculin (V9131) was purchased from Sigma-Aldrich. Secondary anti-rabbit (#170-6515) and anti-mouse (#170-6516) antibodies were purchased from Bio-Rad.

### CRISPR screens

For the dropout screen, A549 cells were screened using a custom sgRNA library targeting human kinases (Wang et al, 2018). Upon generating lentiviral vectors A549 cells were infected at MOI between 0.3 and 0.5, selected with puromycin and a reference sample (t = 0) was collected. Cells were then cultured in presence or absence of 0.35 $\mu$M of indisulam for 10 population doublings, whereas maintaining 1,000× coverage of the library. gRNA sequences were then recovered, amplified, and sequenced to determine the abundance. For sequence depth normalization, a relative total size factor was calculated for each sample by dividing the total counts of each sample by the geometric mean of all totals. All values within a sample were then divided by the respective relative total size factor and rounded off to integer values. A differential analysis between "treated" versus "untreated" condition was performed per sgRNA using DESeq2 (Love et al, 2014). The results of this analysis was used as input for an analysis on the gene level for depletion, using MAGeCK's robust rank algorithm (RRA) (Li et al, 2014) which gives a test statistic, $P$-value and FDR value for enrichm,ent of the sgRNAs of gene towards the top. In addition, we calculated a median $\log_2$ fold change per gene over the sgRNAs based on the DESeq2 output.

For the resistance screen, A549 cells were screened with genome-wide Brunello gRNA library (Doench et al, 2016). Cells were infected and selected as described above, and then cultured in the presence or absence of 3 $\mu$M of indisulam for 3 wk. Data were normalized and analyzed as described above for the dropout screen, except for the RRA analysis which was performed for enrichment instead of depletion. Hits were selected based on FDR smaller or equal to 0.1 and median $\log_2$ fold change. All hits had $\log_2$ fold change greater or equal than 5.

### Plasmids

Single gRNA oligonucleotides were cloned into LentiCRISPR 2.1 plasmid (Evers et al, 2016) by BsmBI (New England Biolabs)

---

formation assay of Panc10.05. Wild-type, control (sgCTRL), and two individual sgCAND1 cells were treated with indicated doses of indisulam for indicated number of d. **(F)** Western blot analysis of RBM39 and CAND1 in Panc10.05 cells. Wild-type, control, and sgCAND1 cells were treated with 0.5 $\mu$M of indisulam for 8 d. GAPDH was used as loading control. **(G)** CUL4-DCAF15 E3 ubiquitin ligase (CRL) complexes get activated by neddylation (N8) which allows ubiquitination of the substrate (RBM39). Neddylation is reversed by NEDD8-activating enzyme (NAE), which can be inhibited by a small molecular inhibitor MLN4924 leading to inactive CRL complex and reduced substrate degradation. **(H)** Western blot analysis of RBM39 and CUL4A in HCT-116 and A549 cells pretreated for 2 h with 1 $\mu$M MLN4924 or 5 $\mu$M MG-132 followed by a 6-h treatment with 0.5 $\mu$M indisulam. GAPDH was used as loading control. The upper CUL4A band (arrow) represents neddylated CUL4A, whereas the lower band represents the deneddylated CUL4A. **(I)** Western blot analysis of RBM39 and CUL4A in HCT-116 (62.5, 125 and 250 nM MLN4924) and A549 (125, 250, and 500 nM MLN4924) cells treated with 0.5 $\mu$M indisulam and increasing doses of MLN4924 for 24 h. The upper CUL4A band (arrow) represents neddylated CUL4A, whereas the lower band represents the deneddylated CUL4A. **(J)** Long-term colony-formation assays of HCT-116 (62.5 nM MLN4924), HCC-1806 (62.5 nM MLN4924), and A549 (125 nM MLN4924) treated with indicated doses of indisulam and a fixed concentration of MLN4924 for 8–13 d depending on the cell line.
Source data are available for this figure.

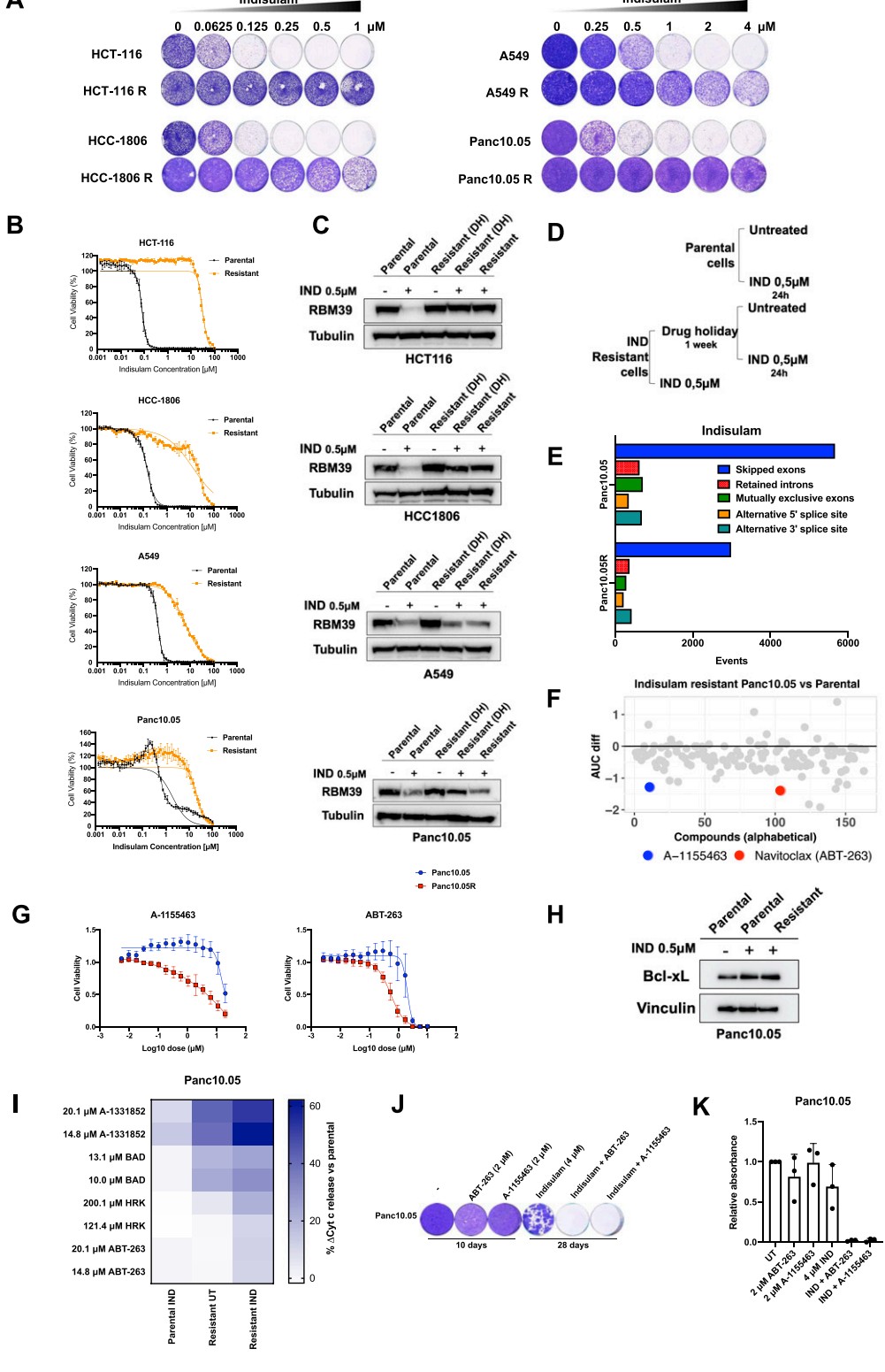

**Figure 4. Cells with acquired resistance to indisulam are vulnerable to BCL-XL inhibition.**
**(A)** Long-term colony-formation assays of HCT-116(R), HCC-1806(R), A549(R), and Panc10.05(R) treated with indicated doses of indisulam for 8–10 d. **(B)** Quantification of cell viability of HCT-116(R), HCC-1806(R), A549(R), and Panc10.05(R) treated with a dilution series of indisulam. Mean of three technical replicates is shown and error bars indicate SD. **(C)** Western blot analysis of RBM39 in parental and resistant HCT-116, HCC-1806, A549, and Panc10.05. Tubulin was used as loading control. **(D)** Experimental design is described in panel (D). **(D)** Parental cells were treated for 24 h with 0.5 μM of indisulam. Resistant cells were cultured without indisulam for 1 wk (drug holiday, DH) and then treated for 24 h with 0.5 μM of indisulam or cultured continuously in the presence of 0.5 μM indisulam. **(E)** Quantification of splicing errors in RNA sequencing data from Panc10.05 cells treated for 18 h with 2 μM indisulam and Panc10.05R cells cultured on 2 μM indisulam. Resistant cells cultured without indisulam for 1 wk were considered untreated. Data were analyzed based on two technical replicates and bars represent the number of events compared with untreated samples. **(F)** Compound screen in resistant and parental Panc10.05 cells. Dose response curves of various compounds were generated. Comparison of area under the curve of parental versus resistant Panc10.05 is plotted for every compound. Compounds validated after a secondary screen are highlighted. **(G)** Cell viability of Panc10.05(R) cells treated with ABT-263 and A-1155463. Indisulam-resistant cells were cultured in the presence of 0.5 μM indisulam. Mean of three biological replicates is shown and error bars indicate SD. **(H)** Western blot analysis of BCL-XL in Panc10.05 parental and resistant cells. Parental cells were treated with 0.5 μM of indisulam for 24 h and resistant cells were cultured in the presence of 0.5 μM indisulam. Vinculin was used as a loading control. **(I)** Heat map of delta cytochrome c release compared with parental untreated cells (%) in Panc10.05(R) cells after BH3 profiling with A-1331852, BAD, HRK, and ABT-263. Before profiling, parental Panc10.05 cells were treated with 0.5 μM of indisulam for 24 h. Resistant Panc10.05 cells were cultured in the absence of indisulam for 2 wk and treated Panc10.05R cells were cultured in the presence of 0.5 μM indisulam. Mean of three technical replicates is shown. **(J)** Long-term colony-formation assay of Panc10.05 cells treated with 2 μM ABT-263, 2 μM A-1155463,

4 μM indisulam, and the combinations for the indicated duration. Representative image of three independent biological replicates is shown. **(K)** Quantification of long-term colony-formation assays of Panc10.05. Mean of three biological replicates is shown and error bars indicate SD. Source data are available for this figure.

digestion followed by Gibson Assembly (New England Biolabs). Control sgRNA: ACGGAGGCTAAGCGTCGCAA, sgRNA targeting *CAND1* #1: AGTCTAGGGCTGGTCAACTG, sgRNA targeting CAND1 #2: AATG-CAATGGATGCTGATGG, sgRNA targeting *SRPK1* #1: GCAACAGAATGGC AGCGATC, sgRNA targeting SRPK1 #2: TGGTAGATCACTCTCAGAGT. The lentiviral shRNA vectors were selected from the arrayed TRC human genome-wide shRNA collection. Control shRNA: CCTAAGGTTAAGTCG CCCTCGCTCGAGCGAGGGCGACTTAACCTTAGG, shRNA targeting *RBM39* #1: GCCGTGAAAGAAAGCGAAGTA, shRNA targeting RBM39 #2: GCTGGA CCTATGAGGCTTTAT.

### Lentiviral transduction

Second generation lentivirus packaging system (psPAX2 [#12260; Addgene], pMD2.G [#12259; Addgene] and pCMV-GFP as transfection control [#11153; Addgene]) was used for lentiviral production. HEK293T cells were transfected using PEI and lentiviral supernatant was then filtered and used to infect target cells using 8 mg/ml Polybrene. Infected cells were then selected with 2 mg/ml puromycin until non-transduced control cells were dead.

### Quantification of editing efficiency

Target sequences were amplified by PCR and SANGER sequenced (Macrogen), then purified by ISOLATE II PCR and Gel Kit (#BIO-52059; Bioline) or the Exo-Cip Rapid PCR Cleanup Kit (New England Biolabs). Gene editing efficiency was analyzed using TIDE analysis software (Brinkman et al, 2014). Each sample was corrected for background by subtracting the editing percentage in cells containing the control gRNA. PCR primers used are as follows: sgCAND1 #1 forward: GATTCCCGGGAGTCAGTTTGG, sgCAND1 #1 reverse: CTGAAA TCCAAAAGGCCGCT, sgCAND1 #2 forward: ATGCACTGGCATTTCCACAA, sgCAND1 #2 reverse: CCTAGCCAAGAGAAAACAAGTGG.

### Compound screen

The library consisted of 164 compounds with anti-cancer properties (Table S1). The active range of every compound was selected based on literature, to set the highest screening concentration in the dilution range. Parental Panc10.05 (400 cells/well) and indisulam-resistant Panc10.05R cells on 0.5 $\mu$M indisulam (500 cells/well) were seeded in 384 well plates using Multidrop Combi (Thermo Fisher Scientific). Cells were treated with the compound library in a 15-point 1:1.8 dilution series for 5 d using the MicroLab Starlet (Hamilton Robotics). Next, cell viability was measured using a resazurin assay on the EnVision plate reader (PerkinElmer). We used phenylarsine oxide (PAO) as a positive control and DMSO as a negative control. For a random concentration per cell line a technical triplicate was taken along to determine the variance. Plate normalization was performed using the normalized percent inhibition method (Boutros et al, 2006), setting values between 0 (for the median of the positive controls) and 1 (for the median of the negative controls). Response curves were fitted with parameters for high level set to 1 and low level set to 0, The Area under the curve (AUC) was calculated as a measure for overall viability. The AUC value of the parental cell line was subtracted from the AUC of the indisulam-resistant cell line. The top 15 compounds in terms of this

difference score were selected for validation. Secondary screen was performed in three biological replicates after which ABT-263 and A1155463 were the only compounds that validated with a substantial difference.

### Dose response and synergy assay

Antagonistic and synergistic interactions of MLN4924 and SPHINX31 with indisulam were determined in 6-d cell viability assays. Cells were seeded in 96-well plates and treated using a HP D300 Digital Dispenser. PAO and DMSO were used as a positive and negative control, respectively. Drugs and medium were refreshed every 2–3 d. Cell viability was measured using resazurin assay on the EnVision plate reader (PerkinElmer). The data were corrected for PAO treated cells and normalized to DMSO treated cells. Drug antagonism and synergy was analyzed using SynergyFinder 2.0 using the Bliss model and viability as the readout (Ianevski et al, 2020). Data are displayed as means of three biological replicates.

### RNA sequencing

For the indisulam and SPHINX31 experiment, A549 cells were treated for 24 h with 0.5 $\mu$M indisulam, 5 $\mu$M SPHINX31, and the combination. For the resistance experiment, Panc10.05 cells were treated for 18 h with 2 $\mu$M indisulam. Resistant Panc10.05 were cultured in the absence of 2 $\mu$M indisulam for 1 wk and treated Panc10.05R cells were continuously cultured in the presence of 2 $\mu$M indisulam. Total RNA was extracted with RNeasy mini kit (Cat. no. 74106; QIAGEN) including a column DNase digestion (Cat. no. 79254; QIAGEN), according to the manufacturer's instructions. Quality and quantity of total RNA was assessed by the 2100 Bioanalyzer using a Nano chip (Agilent). Total RNA samples having RIN > 8 were subjected to library generation. Strand-specific libraries were generated using the TruSeq Stranded mRNA samples preparation kit (RS-122-2101/2; Illumine Inc.) according to the manufacturer's instructions (part #15031047 Rev.E; Illumina). Briefly, polyadenylated RNA from intact total RNA was purified using oligo-dT beads. After purification, the RNA was fragmented, random-primed, and reverse-transcribed using SuperScript II Reverse Transcriptase (part # 18064-014; Invitrogen) with the addition of Actinomycin D. Second strand synthesis was performed using Polymerase I and RNaseH with replacement of dTTP for dUTP. The generated cDNA fragments were 3' end adenylated and ligated to Illumina paired-end sequencing adapters and subsequently amplified by 12 cycles of PCR. The libraries were analyzed on a 2100 Bioanalyzer using a 7500 chip (Agilent), diluted, and pooled equimolar into a multiplex sequencing pool. The libraries were sequenced with paired-end 150-bp reads on a NovaSeq SP (Illumina Inc.).

### Splicing error quantification

The RNA was isolated and sequenced as described above. For the analysis, sequences were demultiplexed and adapter sequences were trimmed from using SeqPurge (Sturm et al, 2016). Trimmed reads were aligned to GRCh38 using Hisat2 (Kim et al, 2019) using the prebuilt genome_snp_tran reference. Splice event detection was performed using rMats version 4.0.2 by comparing the

replicates of the treated groups to the replicates of the untreated group (Shen et al, 2014). rMats events in the different categories were considered significant when the following thresholds were met: having a minimum of 10 reads, an FDR less than 10% and an inclusion-level difference greater than 10%, as described earlier (Wang et al, 2019). For the statistical analysis of different treatments, we assumed that splicing errors occur independently and with a constant rate; the splicing error rate of the combination treatment was then compared with the sum of the splicing error rates of the individual treatments using a Poisson test.

### Long-term colony-formation assays and proliferation assays

For long-term colony-formation assay, the cells were seeded with densities between 10 and 20,000 cells per well, depending on the cell line. Cells were treated with the indicated doses of the drugs which were refreshed every 2–3 d. At the end of the assay, cells were fixed with 2% of formaldehyde (Millipore) in PBS, stained with 0.1% crystal violet (Sigma-Aldrich) in water and scanned. For proliferation assays cells were plated in 96 or 384-well plates with densities between 125 and 1,000 cells per well. The cells were treated the following day using a HP D300 Digital Dispenser and drugs and medium were refreshed every 2–3 d. Plates were incubated at 37°C and images were taken every 4 h using the IncuCyte live cell imaging system. Confluency was calculated to generate growth curves. For apoptosis assay, caspase-3/7 green apoptosis assay reagent (#4440, 1:1,000; Essen Bioscience) was added to each well. Percentage of apoptotic cells was calculated by dividing the caspase-3/7 green signal by the confluence.

### Western blot analysis

Cells were washed with PBS, lysed using RIPA buffer (25 mM Tris–HCl, pH 7.6, 150 mM NaCl, 1% NP-40, 1% sodium deoxycholate, and 0.1% SDS) containing Halt Protease and Phosphatase Inhibitor Single-Use Cocktail (Thermo Fisher Scientific). Loading buffer and reducing agent (both Thermo Fisher Scientific) were added to the samples, which were boiled for 5 min at 95°C and then separated on 4–12% polyacrylamide gradient gels (Invitrogen). After blotting, the PVDF membranes were incubated with primary antibodies diluted to 1:1,000 in 5% BSA. Secondary antibodies were used at 1:10,000 dilution. Immunodetection was conducted using ECL (Bio-Rad) and a Bio-Rad ChemiDoc Imaging System.

### Quantitative RT-PCR

Total RNA extraction was performed using the ISOLATE II RNA mini kit (Bioline) according to the manufacturer's instructions. Next, RNA was reverse transcribed using the SensiFAST cDNA Synthesis Kit (Bioline) according to the manufacturer's protocol. Quantitative PCR analysis was carried out using SYBR green (SensiFast SYBR No-ROX kit) on an Applied Biosystems 7500 Fast Real-Time PCR System (Thermo Fisher Scientific) in technical triplicates. The results were analyzed using the delta-delta Ct method. The sequences of primers used are as follows: RBM39 forward GTCGATGTTAGCTCAGTG CCTC, RBM39 reverse ACGAAGCATATCTTCAGTTATG, RPL13 forward GGCCCAGCAGTACCTGTTTA, RPL13 reverse AGATGGCGGAGGTGCAG.

### BH3 profiling by intracellular staining (iBH3)

BH3 peptides were purchased from New England Peptide: hBIM Acetyl-MRPEIWIAQELRRIGDEFNA-Amide, mBAD Acetyl -LWAAQRY-GRELRRMSDEFEGSFKGL- Amide, HRK-y Acetyl -SSAAQLTAARLKALG-DELHQY- Amide. Corning Black 384 NBS plates were from Corning (#3575). To profile parental and indisulam-resistant Panc10.05 cells, parental Panc10.05 cells were treated for 24 h with 0.5 $\mu$M indisulam, whereas indisulam-resistant Panc10.05R cells were either cultured in the absence (1 wk) or presence of indisulam (0.5 $\mu$M indisulam). Subsequent iBH3 profiling was performed as in Ryan and Letai (2013). In brief, 1 × 10$^4$ cells per 384-well were seeded in a plate containing titrated doses of BIM (100–0.1 $\mu$M), BAD (50–10 $\mu$M), HRK (200–10 $\mu$M), ABT-263 (20–1 $\mu$M), A-1331852, (20–1 $\mu$M), and alamethicin (BML-A150-0005; Enzo) (25 $\mu$M) in a total of 30 $\mu$l MEB buffer (150 mM Mannitol, 10 mM HEPES-KOH [pH 7.5], 150 mM KCl, 1 mM EGTA, 1 mM EDTA, 0.1% BSA, and 5 mM Succinate) + 0.001% wt/vol digitonin. Cells were exposed to the peptides and BH3 mimetics for 50 min at 26°C before cells were fixed using 10 $\mu$l of 4% formaldehyde for 10 min. Subsequently, 10 $\mu$l neutralization buffer (1.7 M Tris base and 1.25 M glycine, pH 9.1) was added to neutralize the formaldehyde and terminate fixation. Afterwards, 10 $\mu$l of CytoC stain buffer (2% Tween20, 10% BSA [wt/vol] in PBS) + 1:400 Alexa Fluor 647 anti-cytochrome c antibody (Cat. no. 612310; BioLegend) + 1:100 DAPI (1 mg/ml, #D3571; Thermo Fisher Scientific) was added, vortexed, and incubated overnight at 4°C in the dark. Flow cytometric acquisition was performed on a BD Fortessa flow cytometer (BD Biosciences) and analyzed using FlowJo (V10.7.1). The gating strategy was set to live single cells positive for DAPI and positive for cytochrome c. Percentage of cytochrome (cyto) c release was calculated as follows:

Data were represented as the mean of technical triplicates, and Δ%Cyto c release is calculated as the %Cyto c release in resistant cells subtracted by their parental counterparts.

## Data Availability

RNA sequencing data are available from Gene Expression Omnibus (accession GSE200280 and GSE200281).

## Supplementary Information

## Acknowledgements

We would like to thank Finn Edwards for help with experiments. We also thank Netherlands Cancer Institute Genomics core facility for their technical support. The work was supported by grants from the European Research Council to R Bernards and the Dutch Cancer Society through the Oncode Institute.

## Author Contributions

Z Pogacar: conceptualization, formal analysis, validation, investigation, visualization, and writing—original draft.

K Groot: conceptualization, formal analysis, validation, investigation, visualization, and writing—original draft.

F Jochems: investigation, methodology, and writing—review and editing.

M Dos Santos Dias: methodology and writing—review and editing.

A Mulero-Sanchez: investigation and writing—review and editing.

B Morris: investigation.

M Roosen: investigation.

L Wardak: validation and investigation.

G De Conti: investigation and writing—review and editing.

A Velds: data curation, software, and formal analysis.

C Lieftink: data curation, software, and formal analysis.

B Thijssen: software, formal analysis, and investigation.

RL Beijersbergen: resources, formal analysis, supervision, funding acquisition, and writing—review and editing.

R Bernards: conceptualization, resources, supervision, funding acquisition, writing—original draft and project administration.

R Leite de Oliveira: conceptualization, data curation, supervision, visualization, writing—original draft and project administration.

## Conflict of Interest Statement

The authors declare that they have no conflict of interest.

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
