## [Reviewer comments · Life Science Alliance]

Life Science Alliance

Genetic and compound screens uncover factors modulating cancer cell response to indisulam

Ziva Pogacar, Kelvin Groot, Fleur Jochems, Matheus Dos Santos Dias, Antonio Mulero-Sánchez, Ben Morris, Mieke Roosen, Leyma Wardak, Giulia De Conti, Arno Velds, Cor Liefink, Bram Thijssen, Roderick Beijersbergen, Rene Bernards, and Rodrigo Leite de Oliveira

DOI: [10.26508/lsa.202101348](https://doi.org/10.26508/lsa.202101348)

Corresponding author(s): *Rodrigo Leite de Oliveira, Amsterdam UMC; Rene Bernards, The Netherlands Cancer Institute*

Review Timeline:

Submission Date:	2021-12-22
Editorial Decision:	2021-12-22
Revision Received:	2022-04-08
Editorial Decision:	2022-04-19
Revision Received:	2022-04-26
Accepted:	2022-04-26

Transaction Report:

Please note that the manuscript was reviewed at Review Commons and these reports were taken into account in the decision-making process at Life Science Alliance.

December 22, 2021

Re: Life Science Alliance manuscript #LSA-2021-01348

Ziva Pogacar
Netherlands Cancer Institute
Plesmanlaan 121
Amsterdam 1066 CX
Netherlands

Dear Dr. Pogacar,

Thank you for submitting your manuscript entitled "Genetic and compound screens uncover factors modulating cancer cell response to indisulam" to Life Science Alliance, along with the reviews obtained from Review Commons. We invite you to re-submit the manuscript, revised according to your Revision Plan.

Thank you for this interesting contribution to Life Science Alliance. We are looking forward to receiving your revised manuscript.

Sincerely,

B. MANUSCRIPT ORGANIZATION AND FORMATTING:

We thank both reviewers for their feedback and constructive comments. We were pleased to read that the reviewers agree our findings are relevant to the field. We hope that you will find that the revised manuscript adequately addresses the reviewer's comments and is suitable for publication in Life Science Alliance.

Reviewer #1 (Evidence, reproducibility and clarity (Required)):

In this study, Pogacar and colleagues set out to identify modulators of indisulam efficacy via a two-pronged, chemical-genetics approach. Indisulam is a small molecule that has been evaluated in human clinical trials but, given a lack of definitive proof of efficacy, has been discarded. It has recently gained a lot of momentum when it has become clear that indisulam acts as a so-called "molecular glue degrader", which induces the degradation of RBM39 by prompting dimerization between RBM39 and the CRL substrate receptor DCAF15. Based on the re-defined mechanism of action, it is reasonable to assume that charting additional genetic factors that influence (i) efficacy, and (ii) resistance acquisition could support efforts to re-purpose indisulam, which is the motivation behind the presented story.

The authors set out to describe the diverse response that is elicited by indisulam treatment over different cancer cell lines, identifying highly responsive as well as intrinsically resistant models. Aiming to map factors that might underpin this varied response, the authors conduct a kinome-focused CRISPR/Cas9 depletion screen to identify knockouts that would synergize with low-dose indisulam treatment in a semi-sensitive cell model. This led to the identification of SRPK1 as a convincing hit, which was further corroborated via the chemical inhibitor SPHINX31. Mechanistically, the observed synergy appears to be tied to an increase in exon skipping, which might be caused by "doubling down" on the splicing machinery by degrading RBM39 on top of inhibiting SRPK1. In addition, the authors also conducted genome-wide screens to identify factors that are required for indisulam efficacy. Here, the authors identified well-defined regulators of CRL4:DCAF15 activity, and focused their efforts on validating the substrate receptor exchange factor CAND1. Finally, the authors address if/how cells would acquire resistance to indisulam in dose-escalation studies, and if the phenotype of acquired resistance would provide associated vulnerabilities that can be exploited via existing small-molecule drugs. This led to the intriguing finding that some cell lines can get indisulam resistance despite a successful RBM39 degradation in response to indisulam treatment. Moreover, these cells appear to be specifically vulnerable to BCL-xL inhibition, which manifests in the finding that dual treatment with indisulam and known BCL-xL inhibitors prevents the emergence of drug resistance in a subset of the assayed cell lines.

In sum, this is an interesting study that touches on various aspects of how cellular effectors can modulate the efficacy of indisulam. While not going into mechanistic depth and detail with either of these, this study certainly warrants publication, pending clarification of the points listed below.

Major points:

1. Since SPHINX31 shows synergy in short term assays, it would be interesting to see if it also delays the onset of resistance acquisition (similar as the BCL-xL inhibitors seem to do)

We agree with the reviewer that this is an interesting experiment to add. We performed the suggested experiment by treating A549, SUM159 and H2122 cells with Sphinx31,

Indisulam and the combination. We observed that the combination treatment indeed prevented the acquisition of resistance in all three cell lines (Figure 2H,I and Supplemental Figure 2E,F).

2. Knockdown efficiency of the RBM39 hairpins should be quantified via Western (1H)
We agree with the reviewer that adding the data on RBM39 protein level would be beneficial. We have performed this experiment and added the results to the figure (Figure 2D).

3. Western Blots in 3C should also contain the untreated condition of the drug resistant subclones

This is an interesting suggestion. We repeated this experiment and added the data on resistant cells after a drug holiday (Figure 4C,D).

Minor points:

1. Page 6, last sentence of the middle paragraph seems to be incomplete

We would like to thank the reviewer for this comment. We have checked the manuscript again for incomplete sentences and corrected as needed.

2. Can the authors comment on which 2 drugs appear to have an even more profound efficacy as the two BCL-xL inhibitors (3E)?

We agree with the reviewer that this is an important detail to further discuss. Those two drugs were Rapamycin and Prexasertib. We decided to not proceed with Rapamycin, as the range of tested concentrations was not optimal, resulting in unclear dose response curves (see figure below, panel A). On the other hand, after validation we did observe a small difference between resistant and parental cells when treated with Prexasertib (figure below, panel B). However, the difference was small, so it likely won't be relevant. We added an explanation to the result section of the revised manuscript (lines 231-235).

Reviewer #1 (Significance (Required)):

This study provides significant advance in our understanding of cellular effectors modulating indisulam resistance. While most of the data that is presented in Figure 2 has been published, the data in Figure 1 and Figure 3 are of high interest in the field. The study would be of even higher interest if some of the findings would be followed up with

more mechanistic depth, but it has sufficient novelty and interest to warrant publication in its current form.

The manuscript should be of interest to a chemical biology or cancer pharmacology audience.

The experience of this reviewer is in targeted protein degradation and chemical genetics.

****Referee Cross-commenting****

Reviewer #2 raises some important points, particularly to part1/figure 1. I agree that the synergy with SRPK1 should be evaluated via an alternative experimental setup that complements the knock out clones.

We agree with the reviewers that the genetic validation should be expanded to include another gRNA sequence. We included another two knock-out clones generated using an independent gRNA (Figure 1D, E). Furthermore, we added a quantitative proliferation assay of clones and control cells treated with indisulam (Figure 1G).

Alternatively, the authors could attempt a rescue experiment where the knockout clones are substituted with SRPK1 cDNA, aiming to show that the hypersensitivity can be reversed.

We would like to thank the reviewer for the suggestion. We believe that the addition of new clones generated by using an independent gRNA and of a quantitative method now strengthens the conclusion of SRPK1 involvement in sensitivity. Furthermore, we added the genetic validation in additional two cell lines.

I also agree that the lack of more relevant models (primary patient samples, organoids, in vivo) dampens the potential impact of this study.

We acknowledge that the suggested models would improve the relevance of this work. We had attempted to validate the combination of Sphinx31 and indisulam in vivo, but unfortunately we observed fast decrease of Sphinx31 concentration in vivo, leading to undetectable plasma concentrations of Sphinx31 2h after the drug administration (see figure below representing the pharmacokinetic study of Sphinx31 administered at 2mg/kg intraperitoneally in 12mice).

Reviewer #2 (Evidence, reproducibility and clarity (Required)):

The paper from Pogacar et al. reports mechanism of cancer resistance to the (so far) ineffective drug Indisulam, by performing a number of high throughput screens in vitro, and by validating their data in cancer cell lines. The paper is divided in three sections 1) identification of pathways synergizing with IND to increase the drug efficacy, and identification of a druggable intrinsic 2) and 3) acquired resistance mechanism. The drug repurposing approach is worthy and the potential impact in oncology quite high. However, the paper's impact is majorly affected by technical flaws, especially in the part 1, by the variability of findings in the models used, especially in part 3, and by lack of in vivo validation of the experiments.

The major concerns regard the section 1. The authors identified SRPK1 as a kinase involved in the resistance to IND, but this result is not completely demonstrated. First, the knockout validation experiments are done with single cell clones obtained from a single gRNA sequence, while these experiments are normally controlled by bulk KO cells obtained from 2 or more non-overlapping gRNAs (like for instance shown for CAND1 in section 2).

We agree with the reviewer that the genetic validation would be improved by inclusion of an independent sgRNA. The reason we performed these experiments using single cell clones and not bulk KO population as for CAND1 is that SRPK1 KO cells are sensitive to indisulam. In a polyclonal population we would therefore expect residual WT cells to proliferate and obscure the effect of KO cells. On the other hand, CAND1 KO cells are resistant to indisulam and therefore the WT cells in the polyclonal population will be counter selected upon indisulam treatment. We agree with the reviewers that the genetic validation should be expanded to include another gRNA sequence. We therefore included two additional knock-out clones generated using an independent gRNA (Figure 1D, E) and show that all the clones show comparable sensitivity to indisulam.

Also, the determination of higher sensitivity of clones by colony formation assays without statistics makes it hard to estimate effects (1E), despite the fact that better techniques to monitor proliferation are available to the study (such as in 1F). Such approach would allow also a determination of the effect of the KO in untreated condition. At present the

KO seems to introduce a marginal sensitization, only, and probably an effect on growth is already present in the absence of IND.

We agree with the reviewer that quantitative assay would improve the conclusion. We performed a quantitative proliferation assay using two different SRPK1 KO clones and control cells and added the data to the revised manuscript (Figure 1G).

Moreover, more cell lines should be used for validation (as also done in section 2 for Cand1), taken from the 1A-B.

It would be valuable to validate the genetic interaction in additional cell lines. We expanded the genetic validation to two additional cell lines: SUM159 and DLD-1 and confirmed that knock-out of SRPK1 sensitizes cells to indisulam. We included two clones for each of the two independent gRNAs and validated the observation using the colony formation assay and quantitative proliferation assay (Figure 1D, F, H and Supplemental Figure 1A, B, C).

Finally, the synergistic combination of SRPK1 inhibitor and IND could as well come from an off-target effect, as also indicated by the same authors, and is not completely ruled out by the experiments in figure 1I (again, here the KD of RBM39 has an effect on its own and no synergism can be concluded from presented data). The lack of adequate statistical evaluation also affects the interpretation of the results in 1J, as well as in other similar experiments. So, this whole section is inconclusive.

We agree that it would be insightful to add quantification of the RBM39 knock down experiment. We performed a quantitative proliferation assay in RBM39 knock-down and control cells and observed a difference in Sphinx31 sensitivity (Figure 2F). In addition, we also performed statistical analysis on the splicing error data to improve the conclusions in this section (Figure 2G).

In the section 2 the results are more clear and many of the flaws above described are absent, although not much effort is made to better characterize the mechanism underlying the loss of CAND1 and resistance, neither we know from section 3 if CAND1 expression is changed in cells made resistant.

We would like to thank the reviewer for this comment. The mechanism of CAND1 mediated resistance to degraders has been previously described (Mayor-Ruiz et al. 2019). It has been shown that loss of CAND1 locks the CRL complex in hyper neddylylated state leading to auto-degradation of substrate receptor. We have attempted to study DCAF15 auto degradation in CAND1 knock-out cells, but unfortunately no specific DCAF15 antibodies are available to allow for this experiment as also reported elsewhere (Hsiehchen et al. 2020). Furthermore, we agree with the reviewer that it would be interesting to investigate CAND1 levels in spontaneously resistant cells. We added the data on CAND1 levels in resistant cells to Supplemental figure 4A.

In the section 3 the findings are potentially promising but are very much limited by high inter-cell variability, so it is unclear how really impactful these results could be.

We agree with the reviewer that the variability between cell lines makes it more difficult to form conclusions. Nevertheless, we believe that the data is promising enough to warrant further investigation.

Minor:

The reason why DDA1 and DCAF15 are not investigated should be better explained in the result section.

We agree that further explanation would be valuable. We included a comment on those two hits in the revised manuscript (lines 170-172).

In some parts, technical details should be more carefully presented. For instance, sentences like ".. increasing the concentration every few passages" is not acceptable and would not allow anyone to repeat exactly the experiment. Methods section should better describe the experiments made.

We agree with the reviewer's comment. We expanded the explanation in the methods section (line 626).

Reviewer #2 (Significance (Required)):

So, overall, the paper explores many aspects in a superficial and unfocused fashion, rather than concentrating on a single target of potential high clinical relevance in molecular depth. Also, the use of better models, like in vivo (and possibly patient-derived), is required for such story to be considered in a high ranking journal.

****Referee Cross-commenting****

Agree on the comments from Rev1.

April 19, 2022

RE: Life Science Alliance Manuscript #LSA-2021-01348R

Dr. Rodrigo Leite de Oliveira
Amsterdam UMC
Cancer Center Amsterdam
De Boelelaan 1117
Amsterdam 1081 HV
Netherlands

Dear Dr. Leite de Oliveira,

Thank you for submitting your revised manuscript entitled "Genetic and compound screens uncover factors modulating cancer cell response to indisulam". We would be happy to publish your paper in Life Science Alliance pending final revisions necessary to meet our formatting guidelines.

- please upload your main and supplementary figures as single files
- please consult our manuscript preparation guidelines <https://www.life-science-alliance.org/manuscript-prep> and make sure your manuscript sections are in the correct order and the section headers are correct
- please use the [10 author names, et al.] format in your references (i.e. limit the author names to the first 10)
- Figure S2B: scale bars are not visible enough. Please make them bigger

A. FINAL FILES:

B. MANUSCRIPT ORGANIZATION AND FORMATTING:

Sincerely,

Reviewer #1 (Comments to the Authors (Required)):

In this revised version of the initial submission, Pogacar and colleagues provided satisfying explanations and additional experimental evidence to cover all points I have raised after carefully reviewing the initial submission. I hence recommend publication.

Reviewer #2 (Comments to the Authors (Required)):

all the issues have been adequately fixed

April 26, 2022

RE: Life Science Alliance Manuscript #LSA-2021-01348RR

Dr. Rodrigo Leite de Oliveira
Amsterdam UMC
Cancer Center Amsterdam
De Boelelaan 1117
Amsterdam 1081 HV
Netherlands

Dear Dr. Leite de Oliveira,

Thank you for submitting your Research Article entitled "Genetic and compound screens uncover factors modulating cancer cell response to indisulam". It is a pleasure to let you know that your manuscript is now accepted for publication in Life Science Alliance. Congratulations on this interesting work.

DISTRIBUTION OF MATERIALS:

Again, congratulations on a very nice paper. I hope you found the review process to be constructive and are pleased with how the manuscript was handled editorially. We look forward to future exciting submissions from your lab.

Sincerely,
